DOI: 10.1038/s41467-018-03322-9　　　**OPEN**

# Atomic visualization of a non-equilibrium sodiation pathway in copper sulfide

Jae Yeol Park [1,2], Sung Joo Kim[1,2], Joon Ha Chang[1,2], Hyeon Kook Seo[1,2], Jeong Yong Lee[1,2] & Jong Min Yuk[1]

Sodium ion batteries have been considered a promising alternative to lithium ion batteries for large-scale energy storage owing to their low cost and high natural abundance. However, the commercialization of this device is hindered by the lack of suitable anodes with an optimized morphology that ensure high capacity and cycling stability of a battery. Here, we not only demonstrate that copper sulfide nanoplates exhibit close-to-theoretical capacity ($\sim$560 mAh $g^{-1}$) and long-term cyclability, but also reveal that their sodiation follows a non-equilibrium reaction route, which involves successive crystallographic tuning. By employing in situ transmission electron microscopy, we examine the atomic structures of four distinct sodiation phases of copper sulfide nanoplates including a metastable phase and discover that the discharge profile of copper sulfide directly reflects the observed phase evolutions. Our work provides detailed insight into the sodiation process of the high-performance intercalation–conversion anode material.

[1] Department of Materials Science & Engineering, Korea Advanced Institute of Science and Technology (KAIST), 291 Daehak-ro, Daejeon 34141, Republic of Korea. [2] Center for Nanomaterials and Chemical Reactions, Institute for Basic Science (IBS), 291 Daehak-ro, Daejeon 34141, Republic of Korea. These authors contributed equally: Jae Yeol Park and Sung Joo Kim. Correspondence and requests for materials should be addressed to J.Y.L. (email: j.y.lee@kaist.ac.kr) or to J.M.Y. (email: jongmin.yuk@kaist.ac.kr)

Since its first discovery in 1980s, lithium ion battery (LIB) has attracted enormous interest from both academia and industry. A combined effort of scientific exploration and industrial optimization has sparked a surge of renewable energy production for a wide range of electronic devices. However, the predicted global depletion of lithium resources before 2025 is alarming enough to trigger a search for another candidate for an energy storage system. In such context, sodium ion battery (SIB) has attracted great attention as a potential alternative to LIB because of its natural abundance and low cost. However, one of the greatest challenges for SIB is to develop a high-performance anode that ensures high capacity and cycling stability of the battery. Metal sulfides such as FeS, Ni$_3$S$_2$, and SnS$_2$ have been widely considered as anode materials because of their higher electrical conductivity and mechanical and thermal stability than the oxides[1–4]. Among them, copper sulfide (CuS) has been considered promising because of its high electrical conductivity (~10$^3$ S cm$^{-1}$) and specific capacity (~560 mAh g$^{-1}$)[5]. Especially, a CuS nanostructure provides a large surface area facilitating Na insertion and extraction, and is thus promising for high capacity and long-term cyclability. Despite its feasibility, the sodiation mechanism of CuS is far from being understood unlike that of lithiation, which undergoes a two-step reaction path from intercalation to displacement[5–7]. Lithium intercalation into CuS initially induces Li$_x$CuS ($0 < x < 1$) formation without any crystallographic reconstruction. Further lithium insertion leads a phase change to Cu$_{2-x}$S (e.g., Cu$_{1.96}$S), followed by formation of a crystalline Li$_2$S matrix and Cu dendrites[5, 6]. These two reactions are directly reflected in the two characteristic plateaus in a voltage profile.

Several reports on CuS sodiation have suggested that Na ions, monovalent like Li, would follow a similar reaction pathway to lithiation[8, 9]. However, the reaction mechanism cannot be considered so simple because the discharge profile of CuS shows more reaction plateaus than lithiation. It has been reported in many intercalation reaction systems that Na insertion experiences a higher diffusion barrier than Li due to the large ionic and atomic radii of Na that introduce large local strain to the host lattice[10–12]. Hence, this affects the reaction kinetics of CuS and thus causes a reaction to deviate from thermodynamic equilibrium.

In this report, we investigate how sodiation proceeds in CuS nanoplates in real time. We visualize the entire process at the atomic scale by employing high-resolution transmission electron microscopy (HR-TEM). This technique allows us to identify multiple distinct phases at various reaction stages and determine the reaction kinetics by quantitative imaging analysis. We support these observations by comparing them with the results from density function theory (DFT) calculation and ex situ experiments. This study, using a high-resolution technique, provides in-depth knowledge on the unique electrochemistry of a SIB electrode.

## Results

**Morphology and electrochemical performance of CuS nanoplates**. Synthesized CuS nanoplates (space group: P6$_3$/mmc) with a unique morphology with two thin interweaving plates have an average diameter of ~300 nm and thickness of ~30 nm (Fig. 1a, b). The plates have a hexagonal shape with crystalline {100} and {001} facets (Fig. 1c, d), and this morphology provides a large surface area for Na insertion and extraction (Supplementary Fig. 1).

To measure electrochemical performance, an electrochemical cell is charged and discharged between 0.05 and 2.6 V because there is only negligible contribution from capacities above 2.6 V and below 0.05 V (Supplementary Fig. 2). The charge/discharge profiles and capacity performance of CuS nanoplates are presented in Fig. 2. After reaching the highest discharge capacity of ~680 mAh g$^{-1}$ at the first cycle, the nanoplates experience severe capacity degradation down to ~80 mAh g$^{-1}$ during the first five cycles. Excess capacity from the first discharge originates from the solid electrolyte interphase (SEI) formation induced by electrolyte decomposition[13]. The sudden capacity drop during the initial five cycles is associated with the rapid loss of Na mobility inside the Na$_x$CuS structure. However, the capacity gradually recovers from the sixth cycle and reaches ~560 mAh g$^{-1}$ at the 100th cycle, which is close to the theoretical capacity of CuS. This recovery can be attributed to an increasing number of Na successfully escaping from the Na$_x$CuS lattice as the nanoplate slowly disintegrates into small parts with short migration paths for Na (Supplementary Fig. 3)[14]. A decrease in bulk diffusivity of Na for initial five cycles matches well with Na mobility loss (Supplementary Fig. 4a). Moreover, the high coulombic efficiency (>100%) for 10−100 cycles (Fig. 2), a drop in charge-transfer resistance ($R_{ct}$), and the enhanced bulk diffusivity from 20 to 100 cycles of the disintegrating nanoplates with cycle numbers agree well with the capacity increase observed upon cycling (Supplementary Fig. 4b).

Once recovered, the capacity and coulombic efficiency are maintained close to ~560 mAh g$^{-1}$ and ~100%, respectively for 50 cycles, and even more (>240 cycles, see Supplementary Fig. 5). Charge and discharge capacities recover well even after cycled at high current densities (Supplementary Fig. 5).

Both intercalation and conversion reactions can be expressed as follows:

$$CuS + xNa^+ + xe^- \leftrightarrow Na_xCuS \tag{1}$$

$$Na_xCuS + (2-x)Na^+ + (2-x)e^- \leftrightarrow Na_2S + Cu \tag{2}$$

where $x$ is the Na content per formula. The presence of more than two plateaus in the discharge profile indicates that more than two steps of intercalation–conversion reaction occurs during sodiation. Hence, we employ in situ TEM to examine the structures of all the reaction phases and understand the sodiation mechanism of CuS.

**Real-time observation of the sodiation process in CuS**. To understand the sodiation dynamics of CuS, real-time observation of CuS nanoplate sodiation is performed in TEM as presented in Fig. 3a and Supplementary Movie 1. A NaF particle is decomposed by electron beam irradiation that generates Na and triggers the chemical reaction between Na and CuS. A reaction front, indicated by a dark cyan line, propagates from left to right upon sodiation. Even before the first front propagation finishes, another front, marked by a purple line, follows the first one. This clearly indicates that two different reactions occur simultaneously. Different diffraction contrasts and patterns of two zones indicate that they are under intercalation (e.g., Na$_x$CuS) and conversion (e.g., Na$_2$S and Cu) reactions, respectively (Supplementary Fig. 6). The conversion reaction produces both a Na$_2$S matrix and Cu nanoparticles (Supplementary Figs. 6d and 7). The observation of a crystalline Na$_2$S matrix suggests that CuS sodiation follows the displacement route rather than the conventional conversion route that produces the amorphous Na$_2$S matrix[8]. The result agrees well with ex situ TEM imaging performed after the standard electrochemical test of CuS nanoplates (Supplementary Figs. 8a and 9).

To quantitatively analyze the dynamics of CuS sodiation, the areal changes of the two reaction regions are measured from time-sequential TEM images (Fig. 3b). Upon Na insertion, the intercalation reaction begins with its area increasing gradually, initially during sodiation. The linear propagation of the first front

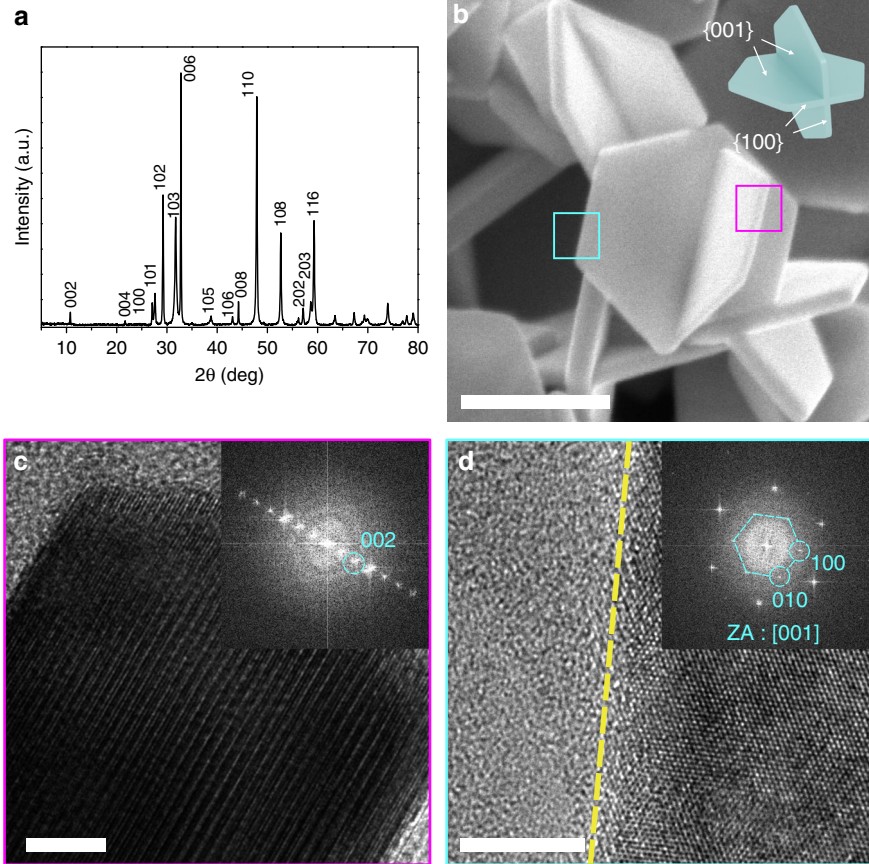

**Fig. 1** Three-dimensional structure of as-synthesized CuS nanoplates. **a** X-ray diffraction (XRD) result of nanoplates. **b** Scanning electron microscopy (SEM) image and the corresponding schematics of three-dimensional CuS nanoplates (scale bar, 200 nm). HR-TEM images of **c** a side plane (scale bar, 5 nm) and **d** a basal plane with [001] zone axis (ZA) showing that each plane corresponds to {100} and {001}, respectively (scale bar, 5 nm)

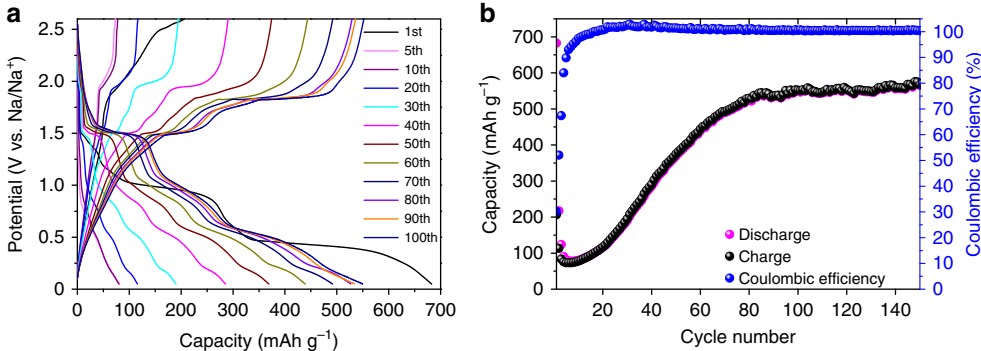

**Fig. 2** Electrochemical performance of CuS nanoplates. **a** Charge and discharge profiles during 100 cycles at 0.2 C between 0.05 and 2.6 V. **b** Graph showing charge and discharge capacity and coulombic efficiency during 150 cycles at 0.2 C

implies that the intercalation reaction is controlled by the reaction rate between Na and CuS (Supplementary Fig. 10); the high mechanical strain induced by large Na insertion controls the intercalation kinetics[15]. Meanwhile, the conversion reaction area also starts to increase soon after. Based on the results from Fig. 3b, the areal change rates from two reactions are obtained (Fig. 3c). As the first front moves away from the Na source, the intercalation reaction dramatically loses its speed, because Na originated from NaF is rapidly consumed by the fast-moving conversion reaction front. Ultimately, the two reaction fronts become indistinguishable from one other, and the entire nanoplate undergoes the conversion reaction.

**HR-TEM observation of $Na_xCuS$ phase evolution.** A discharge profile of CuS presented in Fig. 2a shows more than two plateaus. This implies that CuS nanoplates undergo more than a simple two-step intercalation–conversion process. Hence, we perform real-time HR-TEM to examine the atomic structures of all the reaction phases and fully understand the sodiation mechanism of CuS. The Na intercalation reaction front propagates along the <210> direction followed by the propagation of the conversion reaction front as described in a schematic in Fig. 4a (see Supplementary Movie 2). Like a domino, the propagation is characterized by layer-by-layer insertion of Na in parallel to {100} planes. HR-TEM images are obtained at different stages of

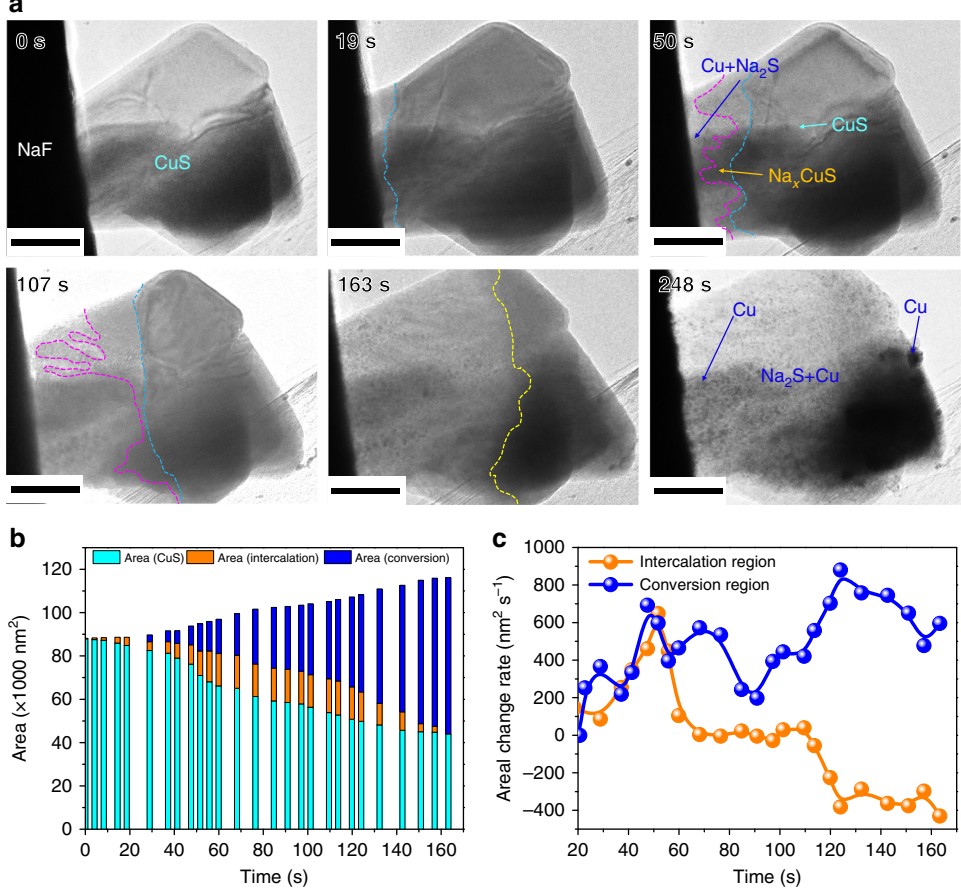

**Fig. 3** In situ observation of the multi-step sodiation process of a CuS nanoplate. **a** Time-sequential bright-field TEM images of a CuS nanoplate showing morphological changes during sodiation (scale bar, 100 nm). Dark cyan and purple lines indicate the intercalation reaction front and the conversion reaction front, respectively. Note that at 163 s, they become indistinguishable (yellow line). **b** Graph of the projected area of each sodiated phase as a function of electron beam irradiation time. Colors used in **a** and **b** represent a pristine phase (CuS, cyan), a phase during the intercalation reaction ($Na_xCuS$, orange), and a phase during the conversion reaction ($Na_2S + Cu$, blue). **c** Graph showing the areal change rates of the intercalation and conversion regions as a function of electron beam irradiation time. The intercalation reaction initiates with its area increasing gradually for the first 57 s. The conversion reaction also initiates at 24 s. From 57 s, the intercalation reaction front loses its speed and its area remains almost unvaried. On the other hand, the conversion reaction has a relatively higher areal increase rate than the intercalation reaction for the whole reaction time. This allows the conversion front to catch up the intercalation front

sodiation and displayed with the corresponding atomic model schematics in Fig. 4 and Supplementary Fig. 11.

Na insertion into CuS generates multiple phases. At the first intercalation step, pristine CuS (Fig. 4b) with a hexagonal structure ($P6_3/mmc$) changes to structurally similar $Na(CuS)_4$ (Fig. 4c) with a trigonal structure ($P\bar{3}m1$), accompanied by a slight expansion of a CuS lattice along $a$ and $b$ axes (Supplementary Fig. 12). Inserted Na atoms break $CuS_x$ tetrahedral columns (marked as blue tetrahedra) and are positioned at {001} planes (marked as yellow green plates in Fig. 4c) in a $Na(CuS)_4$ lattice. Na insertion further expels neighboring Cu atoms and repositions them with S atoms. As a result, Cu and S atoms are coordinated in roto-inversion symmetry along Na planes.

With further Na insertion, the second structural transition occurs. Na atoms are inserted along {001} planes in a $Na(CuS)_4$ crystal and rearrange themselves inside the structure, leading to $CuS_x$ tetrahedron reconstruction. As a result, a new metastable, monoclinic structure ($P2/m$) of a $Na_7(Cu_6S_5)_2$ crystal is formed (Fig. 4d). During this transition, the average bond length between Cu and S atoms increases by ~12%.

At the final intercalation step, the metastable $Na_7(Cu_6S_5)_2$ phase changes to the orthorhombic $Na_3(CuS)_4$ (*Pbam*) phase.

The two structures are crystallographically similar, because of having a {010} $Na_3(CuS)_4$ plane well-matched with a {201} $Na_7(Cu_6S_5)_2$ plane, as shown in a fast-Fourier transform (FFT) pattern in Fig. 4e. During the phase transition to $Na_3(CuS)_4$, lattice expansion of $Na_7(Cu_6S_5)_2$ phase is observed by more Na insertion (Supplementary Fig. 13a, b). Upon further Na insertion, $CuS_x$ tetrahedra translate by 5.16 Å in between the two S rows by breaking the bonds between bridging S atoms (Supplementary Fig. 14). As a result, more Na insertion is allowed between two $CuS_x$ columns while pre-existing Na atoms almost maintain their positions. To verify the presence of all the above intercalation phases during the actual electrochemical cell operation, we conduct the ex situ TEM and X-ray diffraction (XRD) characterization on an electrochemically discharged $Na_xCuS$ nanoplate. HR-TEM image, selected area electron diffraction (SAED) pattern, and XRD data all confirm their presence (Supplementary Figs. 8 and 15, and Supplementary Table 1). More details on above-described phase transitions during the intercalation are described in Supplementary Figs. 11, 12, and 14. FFT patterns of all the phases obtained during the intercalation reaction are displayed in Supplementary Fig. 16, along with simulated patterns based on information from density functional theory (DFT) calculation and

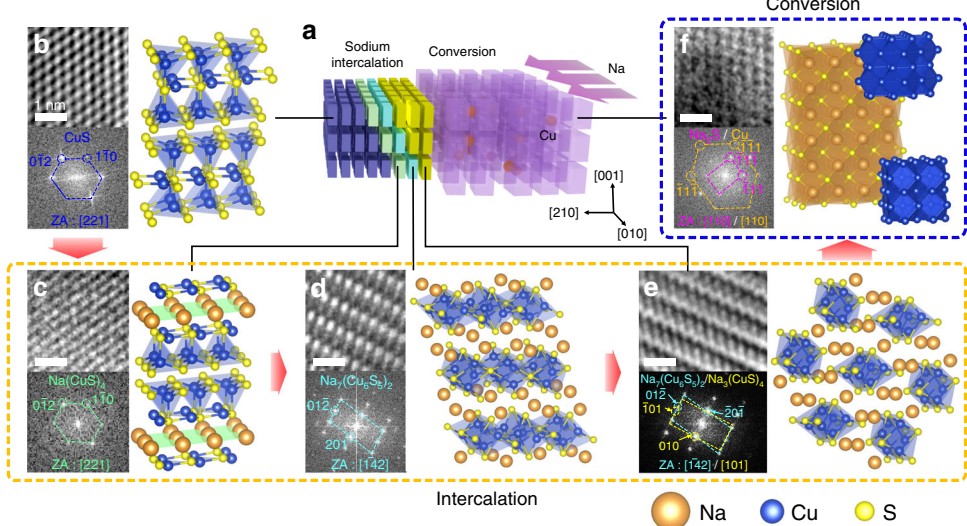

**Fig. 4** HR-TEM observation of CuS nanoplate sodiation. **a** Schematic model demonstrating the entire sodiation process in CuS. Wien-filtered HR-TEM images (scale bar, 1 nm) and the corresponding atomic structures showing **b** CuS, **c** $Na(CuS)_4$, **d** $Na_7(Cu_6S_5)_2$, **e** $Na_3(CuS)_4$, and **f** $Na_2S + Cu$. Blue and orange tetrahedra indicate Cu- and Na-centered tetrahedra, respectively. A yellow green plate corresponds to a plane that consists of inserted Na atoms

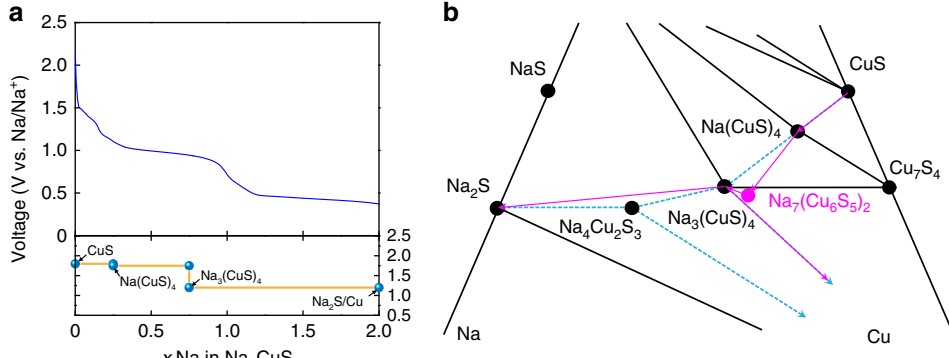

**Fig. 5** Theoretical voltage profile from DFT calculation of CuS. **a** Graph showing the comparison between a first galvanostatic discharge curve (top row of images) at 0.2 C and the DFT calculated voltage profile of a Na–CuS system (bottom row of images). **b** Phase diagram demonstrating the reaction pathway. A dark cyan line indicates a thermodynamic equilibrium pathway predicted by DFT calculation, while a purple line represents a reaction pathway found during in situ experiment. Note the metastable phase, $Na_7(Cu_6S_5)_2$, in the phase diagram

inorganic crystal structure database (ICSD) (Supplementary Tables 2 and 3).

Ultimately, Na insertion over $x_{Na} = 0.75$ transforms the intercalated structure into Cu nanoparticles embedded in a crystalline $Na_2S$ matrix via conversion reaction (Fig. 4f and Supplementary Fig. 17). The four-step sodiation mechanism of CuS nanoplates demonstrated above is structurally reversible as confirmed by the ex situ TEM study after the cell test (Supplementary Fig. 8).

**Electrochemical profile in comparison with DFT calculation**. A calculated voltage profile can be plotted based on Na chemical potentials obtained from Materials Project (Fig. 5b)[16]. The voltage of an electrochemical cell is related to Na chemical potentials according to the following relation:

$$V(x) = -\frac{\mu_{Na}(x) - \mu_{Na}^o}{e} \quad (3)$$

where $\mu_{Na}(x)$ is the Na chemical potential of $Na_x(CuS)_{1-0.5x}$ and $\mu_{Na}^o$ is the Na chemical potential of metallic Na. The thermodynamic equilibrium sodiation pathway determined from a Na–Cu–S ternary phase diagram suggests that each intercalation

and conversion reaction happens in two steps (dark cyan dotted lines in Fig. 5b). However, our in situ TEM observation suggests the presence of three stable phases, $Na(CuS)_4$, $Na_3(CuS)_4$, and $Na_2S+Cu$. This is because there is likely a mechanical barrier for $Na_3(CuS)_4–Na_4Cu_2S_3$ transition due to the lack of structural relationship between two phases. Thus, a three-step route is considered for the DFT calculation. The calculated voltage profile qualitatively matches with the first discharge profile obtained at 0.2 C as shown in Fig. 5a. However, there is still the difference in the voltage profile (e.g., voltage steps) between the experiment and the theoretical calculation because the calculation assumes a bulk system. HR-TEM observation reveals that CuS sodiation is kinetically limited and generates both Na-poor and Na-rich intercalation phases, similar to the miscibility gap, within the nanoplate. Nano-sizing of CuS can change its strain and surface energy, affect the miscibility gap, and alter the voltage profile of CuS by shifting its Gibbs free energy curve[17–19].

## Discussion

Capacity recovery and high capacity retention of CuS nanoplates can be associated with the diglyme-based electrolyte[20]. With a carbonate-based electrolyte (1 M $NaPF_6$ in ethylene carbonate/

dimethyl ethylene carbonate (EC/DEC) with 1:1 volume ratio), a cell experiences severe capacity degradation with low coulombic efficiencies and its capacity is not recovered (Supplementary Fig. 18).

Based on in situ TEM observation, the sodiation pathway of CuS is found quite distinct from that of lithiation. Large Na ion intercalation into $Na_xCuS$ induces strain in the structure and promotes its crystallographic transformation. This mechanism is quite different from lithiation, which generates a solid solution-type transition zone that relieves the misfit strain between the pristine and lithiated phases[5, 7]. The crystallographic tuning forms three intermediate phases and dampens the propagation speed of the intercalation front; this allows the conversion reaction to settle in before the completion of intercalation. Hence, a significant time overlap is observed between the intercalation and conversion reactions like other oxide systems[21, 22].

Crystallographic tuning upon Na intercalation generates the reaction pathway that deviates from the thermodynamic equilibrium. It engenders the evolution of a metastable $Na_7(Cu_6S_5)_2$ phase. Though not thermodynamically favorable, this metastable structure is not only electrochemically active for Na insertion (Supplementary Fig. 8c, e) like that reported in the case of $Na_x(FePO_4)$[23], but also has a structural similarity with both preceding $Na(CuS)_4$ and following $Na_3(CuS)_4$ phases with the coherency strain less than 1%.

The above observation has an important implication to the electrochemical behaviors of a Na–CuS system. As shown in a voltage curve in Fig. 2a, the discharge profile changes continuously with each cycle after the first discharge. Although the voltages of different plateaus remain almost unchanged, the relative capacity contribution for the intercalation and conversion reactions varies throughout the cycle. This hints the possibility of dynamic competition between the intercalation and conversion reactions inside a Na–CuS system with CuS, $Na_xCuS$, and $Na_2S$ + Cu phases co-existing.

In summary, we visualize a non-equilibrium sodiation process of hexagonal CuS nanoplates by employing an in situ TEM technique in conjunction with an electrochemical test and theoretical voltage calculation. We discover that strain induced by large Na insertion into a $Na_xCuS$ lattice promotes crystallographic tuning of the structure and thus generates a reaction pathway that deviates from the thermodynamic equilibrium. The intercalation reaction involves the phase transformation of $Na_xCuS$ into three distinct phases in a following sequence—Na $(CuS)_4$, $Na_7(Cu_6S_5)_2$, and $Na_3(CuS)_4$. Despite being thermodynamically unfavorable, a $Na_7(Cu_6S_5)_2$ phase is generated as a structural bridge between preceding and following stable phases. Our finding is relevant to many other structurally similar anode systems that undergo the intercalation reaction upon sodiation.

## Methods

**CuS nanoplate synthesis and characterization**. Three-dimensional CuS nanoplates are synthesized via a solvothermal method in a Teflon-sealed autoclave as follows[24]. Copper nitrate ($Cu(NO_3)_2 \cdot 3H_2O$), cetyl trimethylammonium bromide (CTAB), hexane, and n-pentanol are purchased from Sigma-Aldrich. Microemulsion, which consists of 0.1 M CTAB (in hexane), 8.65 (molar ratio to CTAB) n-pentanol, and 10 (molar ratio to CTAB) water containing 0.2 M copper nitrate, is used. Copper nitrate aqueous solution is poured into a mixed solution of hexane and pentanol containing CTAB, and they are mixed together under stirring until the solution becomes transparent. The microemulsion is poured into a 100 ml Teflon-sealed autoclave, and 0.8 ml of carbon disulfide is added. The autoclave is transferred into an oven and treated for 15 h at 170 °C. The black precipitate is obtained, washed with acetone and ethanol several times, and dried in a vacuum oven at 60 °C. Scanning electron microscopy (SEM, Varios 460, FEI) and X-ray diffraction (XRD, D/MAX-2500, RIGAKU) are employed to confirm the three-dimensional morphology and the crystal structure of the synthesized nanoplates.

**In situ TEM sample preparation and characterization**. NaF particles and CuS nanoplates are dispersed on a graphene-coated holey carbon Au grid (300 mesh, SPI). TEM (JEM-2100F and ARM-200F, JEOL) equipped with a charge-coupled device (CCD) camera (Orius SC1000, US1000, Gatan) is used for real-time observation of the sodiation process at the accelerating voltage of 200 kV. Sodiation is driven by electron beam irradiation of CuS nanoplates attached to a NaF particle via generation of metallic Na from NaF. During sodiation, time-sequential HR-TEM images and SAED are taken for real-time structural analysis of CuS. To visualize 3D atomic structures of Na intercalated phases, VESTA software is used[25]. After sodiation, energy dispersive spectroscopy (EDS) mapping is also performed to confirm the structural change.

**Electrochemical cell test and ex situ characterization**. To fabricate the working electrode, CuS nanoplates, carbon black (acetylene black, Alfa Aesar), and poly-vinylidene fluoride (PVDF, Sigma-Aldrich) are mixed in 1-methyl-2-pyrrolidone with a weight ratio of 8:1:1. The prepared slurry is coated on a Cu foil and vacuum-dried for 12 h. For the electrolyte, 1 M of sodium hexafluorophosphate ($NaPF_6$) in diglyme is used. The electrolyte is stirred at 80 °C for 48 h inside a glove box under Ar atmosphere[26]. A pure Na foil (Sigma-Aldrich) and a glass fiber (EL-CELL) are used as a reference electrode and a separator, respectively. A half-cell (ECC-STD, EL-CELL) is assembled inside a glove box under Ar atmosphere. The electrochemical cell test is performed using a PARSTAT MC 1000 cell tester (Princeton Applied Research). Charge/discharge profiles are obtained at 0.2 C, at a room temperature between 0.05 and 2.6 V. For ex situ experiments, an electrochemical cell is disassembled after a couple of charge/discharge cycles. For TEM analysis, an active material is thoroughly washed via active sonication for 3 h in dimethyl carbonate (DMC) and dispersed onto a grid for TEM examination. For XRD study, an active material is peeled off and loaded onto a sealed holder to prevent it from contacting with air. The holder is directly equipped inside the XRD machine.

**Data availability**. The data that support the findings of this study are available from the corresponding authors on reasonable request.

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

## Acknowledgements

This work was supported by the Institute for Basic Science (IBS) (IBS-R004-G3), which provided student and postdoc support; Nano·Material Technology Development Program through the National Research Foundation of Korea (NRF) funded by the Ministry of Science, ICT and Future Planning (2009-0082580), which provided support for TEM characterization; and the National Research Foundation of Korea (NRF) grant funded by the Korea government (MSIP; Ministry of Science, ICT & Future Planning) (NRF-2017R1C1B5017962), which provided support for graphene growth and development of graphene transfer method.

## Author contributions

J.Y.P., S.J.K., J.H.C., H.K.S., and J.M.Y. designed the experiments. J.Y.P. synthesized CuS nanoplates and conducted in situ TEM analysis and the electrochemical performance test. S.J.K. performed ex situ TEM analysis and theoretical voltage calculation. J.Y.P. and S.J.K. prepared the manuscript with J.H.C. and H.K.S. under supervision of J.M.Y. and J.Y.L.

## Additional information

**Competing interests:** The authors declare no competing interests.

