## [Peer Review File · Nature Communications]

Reviewers' comments:

Reviewer #1 (Remarks to the Author):

I appreciate authors' efforts in revising the manuscript and believe to certain extend the paper has indeed been improved. However, my main concern on crystal structure determination remains. In my original comments #2, I suggest "the authors include a table, providing detailed crystallographic data including the lattice parameters and space groups as well as the allowed/forbidden reflections for all these phases discussed in this work". Nevertheless, the authors only included a few tables of the atomic positions of the phases under study without listing the intensity, or structure factors (intensity equals to structure factor square, $I=F^2$) of the major reflections. The intensity values of the reflections can be easily calculated based on the atomic positions provided, and they also are available in the x-ray crystallography data base. Without the expected intensity values the reader cannot judge whether these identifications are correct. For instance, looking at the new XRD data Fig. S14 the authors used to respond both referees' comments, how can I tell the indexing of the phases are correct as there are 6 phases for so few peaks. In XRD indexing, it is a common knowledge that indexing a few peaks does not mean the identification of a phase. If the authors can show all the major peaks of the 6 phases are indexed in the XRD data (Fig. S14) I will recommend the paper for publication.

Reviewer #2 (Remarks to the Author):

Park et. al investigated the detailed sodiation process of CuS, and four distinct sodiation phases of CuS nanoplates including a metastable phase were detected, providing a novel sight into the sodiation process of high-performance intercalation-conversion anode material. This work is novel, well organized and result-oriented. However, the following issues should be taken into consideration to improve the manuscript.

1. The phase transitions from CuS to Na(CuS)₄, Na(CuS)₄ to Na₇(Cu₆S₅)₂, Na₇(Cu₆S₅)₂ to Na₃(CuS)₄, and Na₃(CuS)₄ to Na₂S/Cu are confirmed by the HRTEM and SEAD patters. Ex-situ XRD patterns of 1.0 V and 1.4 V discharged are provided, which is not enough. The authors are invited to provide more Ex-situ XRD results to confirm the phase transitions upon sodiation.
2. In Supplementary Figure 1, only one branch can be observed for CuS nanoplate and CuS bulk, respectively. Besides, adsorption and desorption branches should be demonstrated clearly.
3. In Supplementary Figure 2, the author demonstrate that the cathodic peak at 0.09 V corresponds to the phase transition of Na₃(CuS)₄ to Na₂S/Cu. But the electrochemical tests were conducted in a voltage range of 0.5-2.4 V. Indeed, the capacity above 2.4 V can be ignored, but the capacity contribution below 0.5 V can not.
4. The authors demonstrate the sudden capacity drop during initial 5 cycles resulting from the rapid loss of Na mobility inside the Na_xCuS structure, and capacity recovery resulting from the escaping of Na from the Na_xCuS lattice. Detailed characterizations are invited to confirm this point. Besides, the authors should give an explanation for the decreased R_{ct} value upon cycling (Supplementary Figure 4).

Therefore, major revisions are necessary before publication.

Responses to the reviewers' comments/questions:

Reviewers' comments:

Reviewer #1 (Remarks to the Author):

Review report:

“Atomic visualization of a non-equilibrium sodiation pathway in CuS” by Park et al (NCOMMS-17-25300)

I appreciate authors' efforts in revising the manuscript and believe to certain extend the paper has indeed been improved. However, my main concern on crystal structure determination remains. In my original comments #2, I suggest “the authors include a table, providing detailed crystallographic data including the lattice parameters and space groups as well as the allowed/forbidden reflections for all these phases discussed in this work”. Nevertheless, the authors only included a few tables of the atomic positions of the phases under study without listing the intensity, or structure factors (intensity equals to structure factor square, $I=F^2$) of the major reflections. The intensity values of the reflections can be easily calculated based on the atomic positions provided, and they also are available in the x-ray crystallography data base. Without the expected intensity values the reader cannot judge whether these identifications are correct. For instance, looking at the new XRD data Fig. S14 the authors used to respond both referees' comments, how can I tell the indexing of the phases are correct as there are 6 phases for so few peaks. In XRD indexing, it is a common knowledge that indexing a few peaks does not mean the identification of a phase.

If the authors can show all the major peaks of the 6 phases are indexed in the XRD data (Fig. S14) I will recommend the paper for publication.

Response: We really thank the reviewer for the helpful comment. We not only added the XRD peak position and intensity information of all phases in Table 3 based on inorganic crystal structure database (ICSD) but also performed, in addition, an intensive ex-situ XRD study at many voltage levels. As presented in Figs. 1a and S13 and Table 3, the XRD shows all the major peaks of the 6 phases observed during the in-situ TEM experiment. We note that XRD intensity ratio for nanocrystals can be different from that of bulk materials due to both limited number of crystals with each phase and particle size-dependent preferred plane orientation, as reported from a previous study:

[1] Peck, M. A., Langell, M. A. Comparison of Nanoscaled and Bulk NiO Structural and Environmental Characteristics by XRD, XAFS, and XPS. Chem. Mater. 24, 4483-4490 (2012).

Supplementary Figure 15. Ex-situ x-ray diffraction (XRD) patterns of CuS during discharge. (a) A series of XRD patterns taken at various voltages during discharge process. All phases obtained from in-situ TEM are confirmed with XRD. Both calculated and detected XRD peak information are presented in Table 3. Relative intensity ratio can be different from calculated data (bulk) because of particle size-dependent preferred plane orientation of Na_xCuS nanoparticles [1]. A Cu peak is detected at all voltage levels because of the fragments from a copper electrode foil during XRD sample preparation.

Table 3. Calculated X-ray Diffraction information of CuS, $\text{Na}(\text{CuS})_4$, $\text{Na}_7(\text{Cu}_6\text{S}_5)_2$, $\text{Na}_3(\text{CuS})_4$, Na_2S and Cu based on ICSD.

CuS					
2-Theta	d (Å)	Intensity	(hkl)	Detection	
10.807	8.18	8	(0,0,2)	O	
21.74	4.085	1.2	(0,0,4)	O	
27.122	3.285	14	(1,0,0)	O	
27.681	3.22	30	(1,0,1)	O	
29.277	3.048	65	(1,0,2)	O	
31.784	2.813	100	(1,0,3)	O	
32.852	2.724	55	(0,0,6)	O	
38.835	2.317	10	(1,0,5)	O	
43.101	2.097	6	(1,0,6)	O	
44.3	2.043	8	(0,0,8)	O	
47.78	1.902	25	(1,0,7)	O	
47.941	1.896	75	(1,1,0)	O	

52.714	1.735	35	(1,0,8)	O
56.251	1.634	4	(2,0,1)	
57.205	1.609	8	(2,0,2)	O
58.681	1.572	16	(2,0,3)	O
59.345	1.556	35	(1,1,6)	O
63.54	1.463	6	(1,0,10)	
67.306	1.39	6	(1,1,8)	
69.346	1.354	8	(1,0,11)	
69.997	1.343	6	(2,0,7)	
73.995	1.28	10	(2,0,8)	
77.772	1.227	6	(2,1,2)	
79.077	1.21	10	(2,1,3)	
88.915	1.0998	8	(1,0,14)	
89.451	1.0946	10	(3,0,0)	
93.137	1.0607	10	(2,1,8)	
98.668	1.0155	8	(3,0,6)	
Na(CuS) ₄				
2-Theta	d (Å)	I	(hkl)	Detection
7.32	12.074	10.3	(0,0,1)	
14.66	6.037	30.4	(0,0,2)	O
22.07	4.025	1.7	(0,0,3)	
26.86	3.317	9.8	(1,0,0)	O
27.87	3.198	12.2	(0,1,1)	O
29.57	3.019	14.8	(0,0,4)	O
30.73	2.907	100	(0,1,2)	O
35.03	2.56	24	(1,0,3)	O
37.2	2.415	14.6	(0,0,5)	O
40.37	2.232	23.9	(1,0,4)	O
45.01	2.012	5.9	(0,0,6)	
46.48	1.952	13.5	(0,1,5)	
47.44	1.915	59.3	(1,1,0)	
48.07	1.891	0.8	(1,1,1)	
49.92	1.825	6.1	(1,1,2)	
52.91	1.729	0.9	(1,1,3)	
53.2	1.72	12.7	(1,0,6)	
55.35	1.658	1	(2,0,0)	
55.92	1.643	1.1	(2,0,1)	
56.9	1.617	9.4	(1,1,4)	
57.59	1.599	11.2	(2,0,2)	
60.31	1.533	3.6	(0,2,3)	
60.45	1.53	6.1	(0,1,7)	
61.38	1.509	0.3	(0,0,8)	
61.78	1.5	13.3	(1,1,5)	
64.01	1.454	3.9	(0,2,4)	
67.46	1.387	6.6	(1,1,6)	
68.21	1.374	0.4	(0,1,8)	
68.59	1.367	2.8	(2,0,5)	
70.08	1.342	0.1	(0,0,9)	
74.01	1.28	3.8	(0,2,6)	
75.82	1.254	0.5	(2,1,0)	
76.3	1.247	1	(1,2,1)	
76.54	1.244	5.3	(1,0,9)	
77.74	1.227	6.1	(1,2,2)	
79.28	1.207	0.1	(0,0,10)	
80.11	1.197	2.2	(2,1,3)	
80.23	1.195	2	(2,0,7)	
81.06	1.185	0.5	(1,1,8)	
83.41	1.158	2.3	(2,1,4)	

85.52	1.135	1.2	(1,0,10)	
87.27	1.116	0.1	(2,0,8)	
87.63	1.113	2.2	(1,2,5)	
88.33	1.106	5	(3,0,0)	
88.79	1.101	0.1	(0,3,1)	
89.02	1.099	0.1	(1,1,9)	
89.14	1.098	0.2	(0,0,11)	
90.19	1.088	0.6	(0,3,2)	
92.53	1.066	0.3	(0,3,3)	
92.76	1.064	3.6	(2,1,6)	
95.21	1.043	2.7	(0,2,9)	
95.33	1.042	3.1	(1,0,11)	
95.8	1.038	1.5	(3,0,4)	
97.91	1.021	0.1	(1,1,10)	
98.86	1.014	1.6	(1,2,7)	
100.04	1.005	2.2	(0,3,5)	
104.21	0.976	0.6	(0,2,10)	
105.3	0.969	1	(0,3,6)	
106.02	0.964	0.1	(1,2,8)	
106.26	0.963	0.2	(0,1,12)	
107.12	0.957	2.8	(2,2,0)	
107.61	0.955	0.1	(2,2,1)	
107.97	0.952	0.5	(1,1,11)	
109.08	0.946	0.3	(2,2,2)	
111.57	0.932	0.1	(2,2,3)	
112.07	0.929	0.1	(0,0,13)	
113.72	0.92	0.1	(3,1,0)	
114.23	0.917	0.6	(3,1,1)	
114.48	0.916	3.1	(2,1,9)	
114.61	0.915	2.3	(0,2,11)	
115.13	0.913	1	(2,2,4)	
115.77	0.909	1.6	(3,1,2)	
118.39	0.897	0.6	(1,3,3)	
118.92	0.894	0.1	(0,1,13)	
119.46	0.892	0.3	(0,3,8)	
119.72	0.891	1.2	(1,1,12)	
119.86	0.89	1.6	(2,2,5)	
122.18	0.88	0.6	(1,3,4)	
124.69	0.87	0.9	(2,1,10)	
125.98	0.865	0.6	(2,2,6)	
126.55	0.862	0.1	(0,0,14)	
127.28	0.86	0.9	(3,1,5)	
129.06	0.853	0.1	(0,3,9)	
134.05	0.837	2	(1,3,6)	
134.37	0.836	0.7	(1,1,13)	
134.7	0.835	0.3	(1,0,14)	
136.54	0.829	0.1	(4,0,0)	
137.22	0.827	0.3	(0,4,1)	
137.74	0.826	2.8	(2,1,11)	
139.33	0.822	0.6	(0,4,2)	
141.71	0.815	0.1	(3,0,10)	
143.24	0.812	0.9	(3,1,7)	
143.82	0.81	0.1	(2,0,13)	
144.62	0.809	0.2	(2,2,8)	
148.88	0.8	0.2	(4,0,4)	
$\text{Na}_7(\text{Cu}_6\text{S}_5)_2$				
2-Theta	d (Å)	I	(hkl)	Detection
5.447	16.2127	20.3	(0,0,1)	O

5.447	16.2127	20.3	(1,0,0)	O
7.609	11.6096	0.3	($\bar{1}$,0,1)	
7.871	11.2231	0.4	(1,0,1)	
11.006	8.0324	56.1	(2,0,0)	O
12.12	7.2964	13.4	($\bar{1}$,0,2)	
12.12	7.2964	13.4	($\bar{2}$,0,1)	
12.452	7.1025	100	(1,0,2)	O
12.452	7.1025	100	(2,0,1)	O
15.251	5.8048	12.1	($\bar{2}$,0,2)	
15.78	5.6115	3	(2,0,2)	
16.389	5.4042	29	(0,0,3)	O
17.119	5.1754	0.4	($\bar{1}$,0,3)	
17.249	5.1368	0.3	($\bar{3}$,0,1)	
17.476	5.0706	0.1	(1,0,3)	
17.603	5.0343	0.2	(3,0,1)	
19.468	4.5559	1.6	($\bar{3}$,0,2)	O
19.468	4.5559	1.6	($\bar{2}$,0,3)	O
20.095	4.4152	0.4	(2,0,3)	
20.165	4.4001	0.4	(3,0,2)	
21.911	4.0532	0.5	(0,0,4)	
22.115	4.0162	0.1	(4,0,0)	
22.422	3.962	0.3	($\bar{1}$,0,4)	O
22.974	3.8679	0.7	(4,0,1)	O
22.974	3.8679	0.7	($\bar{3}$,0,3)	O
23.101	3.847	0.2	(0,1,0)	
23.764	3.7412	1.1	(1,1,0)	O
23.764	3.7412	1.1	(3,0,3)	O
24.379	3.6482	14.6	($\bar{4}$,0,2)	
24.379	3.6482	14.6	(1,1,1)	
24.919	3.5703	3.8	(2,0,4)	
25.61	3.4755	0.2	(0,1,2)	
25.61	3.4755	0.2	(2,1,0)	
26.166	3.403	1.8	($\bar{1}$,1,2)	
26.166	3.403	1.8	($\bar{2}$,1,1)	
27.116	3.2858	0.1	($\bar{3}$,0,4)	
27.19	3.2771	0.1	($\bar{4}$,0,3)	
27.485	3.2425	0.2	(0,0,5)	
27.743	3.213	1.2	($\bar{2}$,1,2)	
27.743	3.213	1.2	(5,0,0)	
28.1	3.173	2.6	(2,1,2)	
28.1	3.173	2.6	(4,0,3)	
28.238	3.1578	0.8	(1,0,5)	
28.546	3.1243	0.5	(3,1,0)	
28.546	3.1243	0.5	(5,0,1)	
28.974	3.0792	0.2	($\bar{3}$,1,1)	
29.192	3.0567	2.8	(3,1,1)	
29.192	3.0567	2.8	(1,1,3)	
29.329	3.0428	0.5	($\bar{2}$,0,5)	
30.043	2.9721	0.1	(2,0,5)	
30.386	2.9393	0.6	($\bar{3}$,1,2)	
30.386	2.9393	0.6	($\bar{2}$,1,3)	
30.803	2.9005	3.6	(2,1,3)	
30.803	2.9005	3.6	(3,1,2)	
31.869	2.8058	2.8	(4,0,4)	
31.869	2.8058	2.8	($\bar{5}$,0,3)	
32.195	2.7781	11.9	(4,1,0)	
32.546	2.749	6	($\bar{4}$,1,1)	

32.675	2.7384	1.3	(1,1,4)	
32.799	2.7283	2.2	(3,1,3)	
32.799	2.7283	2.2	(4,1,1)	
33.126	2.7021	2.3	(0,0,6)	
33.44	2.6775	1.3	(6,0,0)	
33.44	2.6775	1.3	(1,0,6)	
33.716	2.6562	20	(6,0,1)	O
33.716	2.6562	20	(2,1,4)	O
34.096	2.6274	0.4	(6,0,1)	
34.237	2.6169	1.5	(2,1,4)	
34.636	2.5877	3.2	(2,0,6)	
34.942	2.5657	13.1	(5,0,4)	
34.942	2.5657	13.1	(4,0,5)	
35.376	2.5353	0.5	(2,0,6)	
35.639	2.5171	5.8	(6,0,2)	
35.971	2.4947	5.6	(4,1,3)	
35.971	2.4947	5.6	(3,1,4)	O
36.202	2.4793	27.3	(0,1,5)	O
36.202	2.4793	27.3	(5,0,4)	O
36.404	2.466	3.4	(5,1,0)	
36.497	2.4599	2.2	(1,1,5)	
36.687	2.4476	20.5	(4,1,3)	
36.687	2.4476	20.5	(5,1,1)	
36.984	2.4286	12.6	(5,1,1)	
36.984	2.4286	12.6	(6,0,3)	
37.661	2.3865	2.4	(2,1,5)	O
37.825	2.3765	0.5	(5,1,2)	
38.236	2.3519	4	(2,1,5)	
38.398	2.3424	28.6	(5,1,2)	O
38.836	2.317	10.3	(4,1,4)	O
38.836	2.317	10.3	(0,0,7)	O
39.224	2.295	0.9	(7,0,0)	
39.629	2.2724	9.3	(3,1,5)	
39.629	2.2724	9.3	(6,0,4)	
39.824	2.2617	1.2	(7,0,1)	
40.141	2.2446	0.5	(2,0,7)	O
40.141	2.2446	0.5	(5,0,5)	O
40.452	2.228	1.3	(3,1,5)	
40.452	2.228	1.3	(7,0,2)	
40.571	2.2218	1.7	(5,1,3)	
40.882	2.2056	8.3	(4,0,6)	
40.882	2.2056	8.3	(2,0,7)	
41.015	2.1988	20.1	(1,1,6)	
41.015	2.1988	20.1	(6,1,0)	
41.337	2.1824	0.8	(1,1,6)	
41.59	2.1697	0.3	(6,1,1)	
41.938	2.1525	0.9	(3,0,7)	
42.047	2.1471	0.5	(2,1,6)	
42.223	2.1386	0.8	(7,0,3)	
42.223	2.1386	0.8	(4,1,5)	
42.372	2.1315	0.7	(5,1,4)	
42.677	2.1169	1.2	(2,1,6)	
43.042	2.0998	3.6	(6,0,5)	
43.042	2.0998	3.6	(3,0,7)	
43.41	2.0828	0.5	(5,1,4)	
43.41	2.0828	0.5	(4,1,5)	
43.825	2.0641	7.1	(3,1,6)	

44.012	2.0558	5.1	($\bar{6}$,1,3)	
44.499	2.0344	1.6	(5,0,6)	
44.499	2.0344	1.6	(6,0,5)	
44.737	2.0241	0.1	(3,1,6)	
44.852	2.0192	0.3	($\bar{1}$,0,8)	
44.92	2.0163	0.3	(6,1,3)	
45.252	2.0023	1.4	($\bar{8}$,0,1)	O
45.252	2.0023	1.4	(1,0,8)	O
45.597	1.9879	7.8	($\bar{5}$,1,5)	O
45.597	1.9879	7.8	(8,0,1)	O
45.881	1.9763	2.1	($\bar{1}$,1,7)	
45.881	1.9763	2.1	(4,0,7)	
46.225	1.9624	19.5	($\bar{7}$,1,1)	O
46.225	1.9624	19.5	(1,1,7)	
46.549	1.9494	1.6	(7,1,1)	
46.549	1.9494	1.6	(2,0,8)	
46.821	1.9387	17.3	($\bar{2}$,1,7)	
46.821	1.9387	17.3	(5,1,5)	
47.215	1.9235	32.1	(0,2,0)	
47.477	1.9135	1.5	(2,1,7)	
47.477	1.9135	1.5	(4,1,6)	
47.576	1.9097	2.9	($\bar{5}$,0,7)	
47.576	1.9097	2.9	(6,1,4)	
47.769	1.9025	7.7	(7,1,2)	
47.769	1.9025	7.7	($\bar{8}$,0,3)	
48.418	1.8785	0.6	($\bar{3}$,1,7)	
48.634	1.8706	2.3	(6,0,6)	
48.634	1.8706	2.3	(2,2,0)	
49.025	1.8566	4.5	(1,2,2)	
49.025	1.8566	4.5	(2,2,1)	
49.23	1.8494	2.6	(5,0,7)	
49.403	1.8433	0.6	(3,1,7)	
49.403	1.8433	0.6	($\bar{6}$,1,5)	
49.655	1.8345	2	($\bar{4}$,0,8)	
49.655	1.8345	2	(7,1,3)	
49.906	1.8259	1.3	($\bar{2}$,2,2)	
49.906	1.8259	1.3	($\bar{8}$,0,4)	
50.091	1.8196	0.4	(2,2,2)	
50.311	1.8121	2.4	(3,2,0)	
50.311	1.8121	2.4	(0,2,3)	
50.679	1.7998	6.6	($\bar{4}$,1,7)	
50.679	1.7998	6.6	(5,1,6)	
50.886	1.793	2.6	(0,1,8)	
50.886	1.793	2.6	($\bar{7}$,1,4)	
51.043	1.7879	0.8	($\bar{1}$,1,8)	
51.176	1.7835	3	(6,0,7)	
51.176	1.7835	3	(1,0,9)	
51.31	1.7792	3.9	(8,1,0)	
51.31	1.7792	3.9	($\bar{7}$,0,6)	
51.666	1.7678	2.9	(9,0,1)	
51.79	1.7638	2.9	($\bar{2}$,1,8)	
51.79	1.7638	2.9	(8,1,1)	
52.046	1.7557	5.3	($\bar{9}$,0,2)	
52.384	1.7452	0.2	(2,0,9)	O
52.588	1.7389	0.6	(0,2,4)	
52.588	1.7389	0.6	(2,1,8)	
52.926	1.7286	1.4	($\bar{6}$,1,6)	

52.926	1.7286	1.4	($\bar{1}$,2,4)	
53.135	1.7223	0.2	(4,2,1)	
53.135	1.7223	0.2	(7,0,6)	
53.353	1.7158	0.1	($\bar{3}$,1,8)	
53.47	1.7123	0.1	($\bar{5}$,1,7)	
53.47	1.7123	0.1	($\bar{9}$,0,3)	
53.68	1.7061	0.4	($\bar{8}$,1,3)	O
53.68	1.7061	0.4	($\bar{7}$,1,5)	O
53.836	1.7015	2.2	($\bar{2}$,2,4)	
53.836	1.7015	2.2	($\bar{4}$,2,2)	
54.115	1.6934	0.7	(2,2,4)	O
54.226	1.6902	1.6	(5,0,8)	
54.226	1.6902	1.6	(3,0,9)	
54.403	1.6851	0.5	(3,1,8)	
54.65	1.6781	0.1	(9,0,3)	
54.726	1.6759	0.1	(8,1,3)	
55.052	1.6668	0.1	(5,1,7)	
55.192	1.6629	0.3	(7,1,5)	
55.192	1.6629	0.3	($\bar{4}$,0,9)	
55.502	1.6543	0.1	(0,2,5)	
55.502	1.6543	0.1	($\bar{9}$,0,4)	
55.726	1.6482	0.4	($\bar{8}$,1,4)	
55.726	1.6482	0.4	($\bar{1}$,2,5)	
55.922	1.6429	0.7	($\bar{6}$,0,8)	
55.922	1.6429	0.7	(1,2,5)	
56.35	1.6314	0.3	(0,1,9)	
$\text{Na}_3(\text{CuS})_4$				
2-Theta	d (Å)	I	(hkl)	Detection
12.1	7.31	59	(2,0,0)	O
13.76	6.43	100	(1,1,0)	O
17.31	5.12	2	(2,1,0)	
22.04	4.03	1	(3,1,0)	
24.34	3.654	11	(4,0,0)	
24.84	3.582	2	(0,2,0)	
27.4	3.252	3	(4,1,0)	
27.72	3.216	2	(2,2,0)	
32.5	2.753	11	(3,1,1)	O
33.06	2.707	19	(5,1,0)	
34.14	2.624	7	(4,0,1)	
34.52	2.596	7	(0,2,1)	
35.07	2.557	10	(4,2,0)	
36.47	2.462	18	(4,1,1)	O
36.71	2.446	33	(2,2,1)	
38.15	2.357	9	(1,3,0)	
39.28	2.292	28	(3,2,1)	
39.67	2.27	5	(2,3,0)	
40.99	2.2	4	(5,1,1)	O
42.68	2.117	2	(4,2,1)	
44.21	2.047	6	(6,0,1)	
45.19	2.005	2	(7,1,0)	
45.33	1.999	3	(1,3,1)	
46.08	1.968	4	(6,1,1)	
46.69	1.944	21	(2,3,1)	
48.21	1.886	16	(0,0,2)	O
48.82	1.864	1	(3,3,1)	
49.93	1.825	2	(2,0,2)	
50.4	1.809	5	(1,1,2)	

50.96	1.791	1	(0,4,0)	
51.38	1.777	11	(6,2,1)	
54.56	1.681	4	(3,4,0)	
55.28	1.66	1	(5,3,1)	
55.84	1.645	2	(8,0,1)	
56.48	1.628	2	(8,2,0)	
57.44	1.603	1	(8,1,1)	
59.44	1.554	1	(6,3,1)	
59.72	1.547	4	(5,1,2)	
61.02	1.517	2	(4,2,2)	
62.08	1.494	2	(8,2,1)	
62.22	1.491	4	(6,0,2)	
63.1	1.472	3	(1,3,2)	
63.66	1.461	2	(9,1,1)	
64.12	1.451	5	(7,3,1)	
65.94	1.416	2	(5,4,1)	
66.44	1.406	1	(2,5,0)	
68	1.378	2	(9,2,1)	
69.32	1.355	2	(8,3,1)	
69.72	1.348	2	(6,4,1)	
70.56	1.334	3	(4,5,0)	
Na ₂ S				
2-Theta	d (Å)	I	(hkl)	Detection
23.52	3.78	57.6	(1,1,1)	O
27.22	3.273	4.5	(2,0,0)	
38.88	2.315	100	(2,2,0)	O
45.94	1.974	17.5	(3,1,1)	O
48.1	1.89	1.3	(2,2,2)	
56.15	1.637	12.8	(4,0,0)	
61.71	1.502	5.7	(3,3,1)	
63.49	1.464	1.1	(4,2,0)	
70.39	1.336	20	(4,2,2)	
75.38	1.26	3.6	(5,1,1)	
83.45	1.157	5.2	(4,4,0)	
88.22	1.107	3.2	(5,3,1)	
89.81	1.091	0.1	(6,0,0)	
96.17	1.035	6.3	(6,2,0)	
100.98	0.998	1.1	(5,3,3)	
102.6	0.987	0.1	(6,2,2)	
109.2	0.945	1.5	(4,4,4)	
114.33	0.917	1.8	(5,5,1)	
116.08	0.908	0.1	(6,4,0)	
123.4	0.875	7.2	(6,4,2)	
129.31	0.852	2.4	(7,3,1)	
140.53	0.818	0.8	(8,0,0)	
148.76	0.8	0.7	(7,3,3)	
Cu				
2-Theta	d (Å)	I	(hkl)	Detection
43.2	2.093	100	(1,1,1)	O
50.31	1.812	43	(2,0,0)	O
73.9	1.282	17.7	(2,2,0)	O
89.63	1.093	16.2	(3,1,1)	
94.81	1.046	4.4	(2,2,2)	
116.44	0.906	1.9	(4,0,0)	
135.74	0.832	5.9	(3,3,1)	
143.76	0.81	5.6	(4,2,0)	

Reviewer #2 (Remarks to the Author):

Park et. al investigated the detailed sodiation process of CuS, and four distinct sodiation phases of CuS nanoplates including a metastable phase were detected, providing a novel sight into the sodiation process of high-performance intercalation-conversion anode material. This work is novel, well organized and result-oriented. However, the following issues should be taken into consideration to improve the manuscript.

Detailed comments:

1. The phase transitions from CuS to Na(CuS)₄, Na(CuS)₄ to Na₇(Cu₆S₅)₂, Na₇(Cu₆S₅)₂ to Na₃(CuS)₄, and Na₃(CuS)₄ to Na₂S/Cu are confirmed by the HRTEM and SEAD patters. Ex-situ XRD patterns of 1.0 V and 1.4 V discharged are provided, which is not enough. The authors are invited to provide more Ex-situ XRD results to confirm the phase transitions upon sodiation.

Response 1: We really thank the reviewer for the helpful comment. We performed an additional ex-situ XRD study at more voltage levels and confirmed all phases observed during the in-situ TEM experiment. Since the XRD study shows the change in the average state of all particles and the in-situ TEM study shows the change in the local state of individual particles, both ex-situ XRD and in-situ TEM data are strongly needed to support multiple phase transitions of CuS sodiation.

Supplementary Figure 15. Ex-situ x-ray diffraction (XRD) patterns of CuS during discharge. (a) A series of XRD patterns taken at various voltages during discharge process. All phases obtained from in-situ TEM are confirmed with XRD. Both calculated and detected XRD peak information are presented in Table 3. Relative intensity ratio can be different from calculated data (bulk) because of particle size-dependent preferred plane orientation of Na_xCuS nanoparticles [1]. A Cu peak is detected at all voltage levels because of the fragments from a copper electrode foil during XRD sample preparation.

2. In Supplementary Figure 1, only one branch can be observed for CuS nanoplate and CuS bulk, respectively. Besides, adsorption and desorption branches should be demonstrated clearly.

Response 2: We performed an additional BET experiment on CuS nanoplates and microparticles. As a result, we found nanoplates have a higher surface area than microparticles. They nearly do not have pores based on the adsorption and desorption profiles in relative pressure between 0.5 and 1.0.

Supplementary Figure 1. Brunauer–Emmett–Teller (BET) measurement of CuS nanoplates and bulk. CuS nanoplates have ~2.26 times higher surface area than the CuS bulk. Both nanoplates and bulk have no pores since adsorption and desorption branches are almost same in relative pressure between 0.5 and 1.0.

3. In Supplementary Figure 2, the author demonstrate that the cathodic peak at 0.09 V corresponds to the phase transition of $\text{Na}_3(\text{CuS})_4$ to $\text{Na}_2\text{S}/\text{Cu}$. But the electrochemical tests were conducted in a voltage range of 0.5-2.4 V. Indeed, the capacity above 2.4 V can be ignored, but the capacity contribution below 0.5 V can not.

Response 3: We think that the reviewer might have missed a caption of Fig. S2. We indicated that the electrochemical experiment was performed between 0.05 V and 2.4 V, which covers all electrochemical reaction regimes.

Supplementary Figure 5. Electrochemical performance of CuS nanoplates for >150 cycles. The cell is operated between 0.05 V and 2.4 V until ~ 190 cycles, followed by C-rate dependent capacity measurement between 0.05 V and 2.0 V.

4. The authors demonstrate the sudden capacity drop during initial 5 cycles resulting from the rapid loss of Na mobility inside the Na_xCuS structure, and capacity recovery resulting from the escaping of Na from the Na_xCuS lattice. Detailed characterizations are invited to confirm this point. Besides, the authors should give an explanation for the decreased R_{ct} value upon cycling (Supplementary Figure 4).

Response 4: We performed an additional EIS experiment to investigate the sudden capacity drop during first 5 cycles. From the experiment, we found a slope in a low frequency region in the Nyquist plot decreases upon cycling. This means that bulk diffusivity decreases during initial 5 cycles, implying the loss of Na mobility inside the CuS nanoplate because CuS nanoplates were not disintegrated enough. After initial ~ 10 cycles, coulombic efficiency retains slightly larger than 100% for subsequent ~ 90 cycles (Fig. 2 and Fig. S5) along with bulk diffusivity enhancement from 20th cycle to 100th cycle. This means that trapped Na escapes successfully and, as a result, the capacity gradually recovers. A similar phenomenon has already been reported in ref. [14].

[14] Du, Y., Yin, Z., Zhu, J., Huang, X., Wu, X.-J., Zeng, Z., Yan, Q., Zhang, H., A general method for the large-scale synthesis of uniform ultrathin metal sulphide nanocrystals. *Nat. Commun.* **3**, 1177 (2012).

We added the following statement in our manuscript (page 5, line 10):

“A decrease in bulk diffusivity of Na for initial 5 cycles matches well with Na mobility loss (Supplementary Fig. 4a). Moreover, high coulombic efficiency (>100%) for 10 ~ 100 cycles (Fig. 2), a drop in charge-transfer resistance (R_{ct}) and enhanced bulk diffusivity from 20 to 100 cycles of the disintegrating nanoplates with cycle numbers agree well with the capacity increase happened upon cycling (Supplementary Fig. 4b).”

For decreased R_{ct} , we have explained that it is related to the disintegration of CuS nanoplates during cycling, which increases an exposed surface area for facile Na insertion and extraction into CuS.

Supplementary Figure 4. Nyquist plots from electrochemical impedance spectroscopy (EIS) results upon cycling. The EIS performed (a) after 1st, 3rd, and 5th and (b) after 20th, 100th and 150th discharges within the frequency range between 10 kHz and 0.1 Hz at the amplitude of 10 mV. In the EIS, profiles in high-medium frequency are attributed to charge-transfer resistance (R_{ct}) while low frequency regions are associated with bulk solid-state diffusion of Na ions. For the first 5 cycles, both R_{ct} and the slope of low frequency regions decrease. The decrease of R_{ct} can be associated with CuS nanoplate fracturing into nano-sized grains upon cycling. The decline of the slope is associated with the loss of Na mobility inside CuS nanoplates because CuS nanoplates are not yet fully disintegrated (see inset images of Supplementary Fig. 8a,b). For the 20th, 100th and 150th cycles, R_{ct} still decreases while the slope increases from 20 to 100 cycles. This implies enhancement of Na mobility with structural disintegration of CuS nanoplates. The disintegration increases the exposed surface area of CuS, reduces its R_{ct} , and thus facilitates insertion and extraction of Na ions into CuS.

REVIEWERS' COMMENTS:

Reviewer #1 (Remarks to the Author):

In responding to my comments the authors have now added the XRD peak position and intensity information of all six phases in Table 3 based on the inorganic crystal structure database (ICSD). The results from the XRD indexing seem reasonable, although not great as a few major peaks were not identified. For instance, $\text{Na}(\text{CuS})_4$ (110) ($I=59$), $\text{Na}_7(\text{Cu}_6\text{S}_5)_2$ (020) ($I=32$), and $\text{Na}_3(\text{CuS})_4$ (221) ($I=33$). I thus have no problem to accept the paper for publication after the authors shorten the table by removing all the minor reflection peaks from the table. I think the reader will be able to judge the soundness of the work.

Reviewer #2 (Remarks to the Author):

Since all the issues raised have been addressed, the revised version can be published.

Responses to the reviewers' comments/questions:

REVIEWERS' COMMENTS:

Reviewer #1 (Remarks to the Author):

In responding to my comments the authors have now added the XRD peak position and intensity information of all six phases in Table 3 based on the inorganic crystal structure database (ICSD). The results from the XRD indexing seem reasonable, although not great as a few major peaks were not identified. For instance, Na(CuS)₄ (110) (I=59), Na₇(Cu₆S₅)₂ (020) (I=32), and Na₃(CuS)₄ (221) (I=33). **I thus have no problem to accept the paper for publication after the authors shorten the table by removing all the minor reflection peaks from the table. I think the reader will be able to judge the soundness of the work.**

Response: We really thank the reviewer for the comment. Following the comment, we modified the table by removing all minor peaks to make the table more concise for the reader.

CuS				
2-Theta	d (Å)	Intensity	(hkl)	Detection
27.122	3.285	14	(1,0,0)	O
27.681	3.22	30	(1,0,1)	O
29.277	3.048	65	(1,0,2)	O
31.784	2.813	100	(1,0,3)	O
32.852	2.724	55	(0,0,6)	O
38.835	2.317	10	(1,0,5)	O
47.941	1.896	75	(1,1,0)	O
52.714	1.735	35	(1,0,8)	O
58.681	1.572	16	(2,0,3)	O
59.345	1.556	35	(1,1,6)	O
Na(CuS) ₄				
2-Theta	d (Å)	Intensity	(hkl)	Detection
7.32	12.074	10.3	(0,0,1)	
14.66	6.037	30.4	(0,0,2)	O
26.86	3.317	9.8	(1,0,0)	O
27.87	3.198	12.2	(0,1,1)	O
29.57	3.019	14.8	(0,0,4)	O
30.73	2.907	100	(0,1,2)	O
35.03	2.56	24	(1,0,3)	O
37.2	2.415	14.6	(0,0,5)	O
40.37	2.232	23.9	(1,0,4)	O
46.48	1.952	13.5	(0,1,5)	
47.44	1.915	59.3	(1,1,0)	
53.2	1.72	12.7	(1,0,6)	
57.59	1.599	11.2	(2,0,2)	
Na ₇ (Cu ₆ S ₅) ₂				
2-Theta	d (Å)	Intensity	(hkl)	Detection
5.447	16.2127	20.3	(0,0,1)	O
5.447	16.2127	20.3	(1,0,0)	O
11.006	8.0324	56.1	(2,0,0)	O
12.12	7.2964	13.4	($\bar{1}$,0,2)	
12.12	7.2964	13.4	(2,0,1)	
12.452	7.1025	100	(1,0,2)	O
12.452	7.1025	100	(2,0,1)	O
15.251	5.8048	12.1	($\bar{2}$,0,2)	
16.389	5.4042	29	(0,0,3)	O

24.379	3.6482	14.6	($\bar{4}$,0,2)	
24.379	3.6482	14.6	(1,1,1)	
32.195	2.7781	11.9	(4,1,0)	
33.716	2.6562	20	($\bar{6}$,0,1)	O
33.716	2.6562	20	($\bar{2}$,1,4)	O
34.942	2.5657	13.1	($\bar{5}$,0,4)	
34.942	2.5657	13.1	(4,0,5)	
35.971	2.4947	5.6	($\bar{3}$,1,4)	O
36.202	2.4793	27.3	(0,1,5)	O
36.202	2.4793	27.3	(5,0,4)	O
36.687	2.4476	20.5	(4,1,3)	
36.687	2.4476	20.5	($\bar{5}$,1,1)	
36.984	2.4286	12.6	(5,1,1)	
36.984	2.4286	12.6	(6,0,3)	
38.398	2.3424	28.6	(5,1,2)	O
38.836	2.317	10.3	($\bar{4}$,1,4)	O
38.836	2.317	10.3	(0,0,7)	O
41.015	2.1988	20.1	($\bar{1}$,1,6)	
41.015	2.1988	20.1	(6,1,0)	
46.225	1.9624	19.5	($\bar{7}$,1,1)	O
46.225	1.9624	19.5	(1,1,7)	O
46.821	1.9387	17.3	($\bar{2}$,1,7)	
46.821	1.9387	17.3	(5,1,5)	
47.215	1.9235	32.1	(0,2,0)	
Na₃(CuS)₄				
2-Theta	d (Å)	Intensity	(hkl)	Detection
12.1	7.31	59	(2,0,0)	O
13.76	6.43	100	(1,1,0)	O
24.34	3.654	11	(4,0,0)	
32.5	2.753	11	(3,1,1)	O
33.06	2.707	19	(5,1,0)	
36.47	2.462	18	(4,1,1)	O
36.71	2.446	33	(2,2,1)	
39.28	2.292	28	(3,2,1)	
46.69	1.944	21	(2,3,1)	
48.21	1.886	16	(0,0,2)	O
51.38	1.777	11	(6,2,1)	
Na₂S				
2-Theta	d (Å)	Intensity	(hkl)	Detection
23.52	3.78	57.6	(1,1,1)	O
27.22	3.273	4.5	(2,0,0)	
38.88	2.315	100	(2,2,0)	O
45.94	1.974	17.5	(3,1,1)	O
Cu				
2-Theta	d (Å)	Intensity	(hkl)	Detection
43.2	2.093	100	(1,1,1)	O
50.31	1.812	43	(2,0,0)	O
73.9	1.282	17.7	(2,2,0)	O

Supplementary Table 1. Major peaks from Calculated XRD information of CuS, Na(CuS)₄, Na₇(Cu₆S₅)₂, Na₃(CuS)₄, Na₂S and Cu based on ICSD.

Reviewer #2 (Remarks to the Author):

Since all the issues raised have been addressed, the revised version can be published.

Response: we really thank the reviewer for the final comment.